# Do Airports Have Their Own Climate?

**William A. Gough * and Andrew C. W. Leung** 

Department of Physical and Environmental Sciences, University of Toronto Scarborough,
Toronto, ON M1C 1A4, Canada; andrewc.leung@utoronto.ca
* Correspondence: william.gough@utoronto.ca

**Abstract:** Sixty-four airport climate records were examined across Canada. Day-to-day (DTD) temperature variability metrics were used to assess the nature of the local environment. In total, 86% of the airports were assessed as peri-urban, reflective of either their location at the fringe of the urban centers or the creation of a peri-urban microclimate by the airport itself. The remaining nine stations were identified using a previously identified metric as marine, or "mountain", a new category developed in this study. The analysis included a proposal for a decision flow chart to identify the nature of the local climate based on DTD thermal variability. An analysis of the peri-urban thermal metric and population indicated that a peri-urban climate was consistently identified for airports independent of the magnitude of the local population (or urbanization), lending support to the idea of a localized "airport" climate that matched peri-urban characteristics.

**Keywords:** urban; marine; rural; mountain; peri-urban; day-to-day temperature variability; climate classification; aviation

## 1. Introduction

Due to an intrinsic need for weather data for aviation purposes, weather stations have been installed at airports around the world for many decades. The data from these stations are generally considered to be of high quality due to the airport's demand for timely and accurate measurements of different atmospheric variables. Ideal sensor locations at airport weather stations are well-ventilated, well-exposed to the environment, uninfluenced by nearby obstacles such as high-rise buildings or trees, and well-situated on an even surface with natural and low vegetation. In addition, the stations are typically stationary and not prone to relocation.

While typical weather stations measure only temperature and precipitation, airport weather stations often measure additional variables such as snow depth, pressure, wind conditions, cloud height and type, visibility, and present weather conditions, as they present possible hazards and disruptions to aircraft and airport operations [1,2]. Some airports measure other supplementary climate variables such as soil temperature, evaporation rate, and solar radiation level [3]. While crowdsourced atmospheric data in urban environments are also available, the data quality is often suspect. In Berlin, Germany, a study comparing Netamos personal weather stations with the reference urban weather stations found that poor metadata, sensor failure, and poor sensor siting led to a reduction of 53% of total data availability [4]. Another study conducted on Netamos stations in Amsterdam, the Netherlands, found that wind speed was systematically underestimated and required additional bias correction before the data could become usable [5]. Observations recorded by smartphones were also unreliable. One type of sensor installed in smartphones took in excess of 15 min to obtain accurate temperature and relative humidity readings [6]. These examples highlight that airport weather stations remain an ideal choice for climatological analysis.

While citizens generally do not live in the vicinities of the airport due to excessive aircraft noise, airport data often serve as a proxy for local urban areas because its characteristics (lack of vegetation, large areas of impermeable surface, and rapid drainage of

precipitation) mirror those of an urban environment for urban heat island research [7,8]. Given the quality of the airport data and its use to represent more generally greater geographic regions, it is important to assess the nature of this high-quality data source and potential issues with airport location and the impact of the airport itself on climate data produced. Using a day-to-day (DTD) temperature variability framework [9], we identified a thermal metric that identified peri-urban environments. Although not exclusively, many of the locations identified as peri-urban were airport weather stations, somewhat unsurprising as airports are typically located on the edge of urban centers and have a different urban heat island signature than the downtown core [7,9,10]. A longitudinal analysis of the University of Toronto climate data located at the center of an urban area illustrated the succession from rural, peri-urban, and urban, reflective of the history of the University which has not changed location but was gradually surrounded by urbanization, while being unconnected to an airport [9]. The question though remains whether the peri-urban nature of airport locations is a result of the location at the urban fringe or a specialized "airport climate", or both.

The DTD temperature variability framework [11] has been used to identify a range of environments including urban and rural [12,13], coastal [14–16], and peri-urban [9]. The urban/rural differentiation used the DTD temperature difference between the maximum temperature of the day and the minimum temperature of the day. The coastal metric relied only on variations of the minimum temperature of the day. The peri-urban metric used a ratio of warm to cold DTD transitions. For peri-urban environments, the magnitude of the warm transitions was larger than cold transitions.

Given the dominance of airport stations in the peri-urban category found in [9], this study explored this behaviour for a larger number of airport climate data sets. This current study has been expanded to include 64 airports.

## 2. Materials and Methods

Sixty-four climate records from Canadian airports were used as listed in Table 1 and depicted in Figure 1. These were obtained from the Canadian climate data archive of Environment and Climate Change Canada (https://climate.weather.gc.ca/) (accessed on 8 February 2022). For most stations, the years 1991–2000 were used, consistent with [9,15]. Some departures from this occurred due to data availability and quality, as listed in Table 1. The population of the local urban area was obtained from the 2001 Canadian census and served as a proxy variable for the magnitude of urbanization [17].

**Table 1.** List of airport climate stations. Numbers match those in Figure 1. Stations are ordered by longitude. Elements included are latitude, longitude, elevation, local population, and data range.

| Station # | Station Name | Latitude (N) | Longitude (W) | Elev. (m) | Population | Data Range |
|---|---|---|---|---|---|---|
| 1 | Whitehorse A | 60.71 | 135.08 | 706.2 | 16,843 | 1998–2007 |
| 2 | Prince Rupert A | 54.29 | 130.44 | 35.4 | 14,643 | 1991–2000 |
| 3 | Terrace A | 54.47 | 128.58 | 217.3 | 16,795 | 1994–2003 |
| 4 | Port Hardy A | 50.68 | 127.37 | 21.6 | 4608 | 1991–2000 |
| 5 | Campbell River A | 49.95 | 125.27 | 108.8 | 31,294 | 1991–2000 |
| 6 | Comox A | 49.72 | 124.9 | 25.6 | 14,028 | 1991–2000 |
| 7 | Nanaimo A | 49.05 | 123.87 | 28 | 77,845 | 1991–2000 |
| 8 | Victoria A | 48.65 | 123.43 | 19.5 | 288,346 | 1991–2000 |
| 9 | Vancouver Int'l A | 49.2 | 123.18 | 4.3 | 1,829,854 | 1991–2000 |
| 10 | Prince George A | 53.89 | 122.68 | 691.3 | 66,239 | 1991–2000 |

**Table 1.** *Cont.*

| Station # | Station Name | Latitude (N) | Longitude (W) | Elev. (m) | Population | Data Range |
|---|---|---|---|---|---|---|
| 11 | Abbotsford A | 49.03 | 122.36 | 59.1 | 129,475 | 1991–2000 |
| 12 | Clinton A | 51.14 | 121.5 | 1126.2 | 641 | 1995–2004 |
| 13 | Dawson Creek A | 55.74 | 120.8 | 654.7 | 10,754 | 1991–2000 |
| 14 | Kamloops A | 50.7 | 120.44 | 345.3 | 67,952 | 1994–2003 |
| 15 | Penticton A | 49.46 | 119.6 | 344.4 | 34,686 | 1995–2004 |
| 16 | Kelowna A | 49.96 | 119.38 | 429.5 | 108,330 | 1991–2000 |
| 17 | Blue River A | 52.13 | 119.29 | 690.4 | 157 | 1991–2000 |
| 18 | Golden A | 51.3 | 116.98 | 784.1 | 4020 | 1991–2000 |
| 19 | Cranbrook A | 49.61 | 115.78 | 940 | 18,528 | 1991–2000 |
| 20 | Yellowknife A | 62.46 | 114.44 | 205.7 | 16,055 | 1991–2000 |
| 21 | Calgary A | 51.12 | 114.02 | 1084.1 | 879,277 | 1991–2000 |
| 22 | Edmonton Int'l A | 53.31 | 113.58 | 723.3 | 782,101 | 1994–2003 |
| 23 | Lethbridge A | 49.63 | 112.8 | 928.7 | 67,374 | 1991–2000 |
| 24 | Cold Lake A | 54.42 | 110.28 | 541 | 14,961 | 1994–2003 |
| 25 | Saskatoon A | 52.17 | 106.72 | 504.1 | 196,816 | 1991–2000 |
| 26 | Brandon A | 49.91 | 99.95 | 409.4 | 39,716 | 1991–2000 |
| 27 | Thunder Bay A | 48.37 | 89.32 | 199 | 103,215 | 1991–2000 |
| 28 | Wawa A | 44.78 | 84.78 | 287.1 | 3279 | 1991–2000 |
| 29 | Sault Ste Marie A | 46.48 | 84.5 | 192 | 67,385 | 1991–2000 |
| 30 | London A | 43.03 | 81.15 | 278 | 377,316 | 1991–2000 |
| 31 | Sudbury A | 46.63 | 80.8 | 348.4 | 103,879 | 1991–2000 |
| 32 | Moosonee A | 51.29 | 80.61 | 9.1 | 1481 | 1996–2005 |
| 33 | Hamilton A | 43.17 | 79.93 | 237.7 | 816,820 | 1991–2000 |
| 34 | Toronto Pearson Int'l A | 43.68 | 79.63 | 173.4 | 4,366,508 | 1991–2000 |
| 35 | North Bay A | 46.36 | 79.42 | 370.3 | 51,895 | 1991–2000 |
| 36 | Toronto Island A | 43.63 | 79.4 | 76.5 | 4,366,508 | 1991–2000 |
| 37 | Toronto Buttonville A | 43.86 | 79.37 | 198.1 | 4,366,508 | 1991–2000 |
| 38 | Peterborough A | 44.23 | 78.37 | 191.4 | 73,303 | 1991–2000 |
| 39 | Trenton A | 44.12 | 77.53 | 86.3 | 43,577 | 1991–2000 |
| 40 | Petawawa A | 45.97 | 77.32 | 130.1 | 10,656 | 1991–2000 |
| 41 | Ottawa A | 45.32 | 75.67 | 114 | 827,854 | 1991–2000 |
| 42 | Montreal Mirabel Int'l A | 45.67 | 74.03 | 82.6 | 50,513 | 1991–2000 |
| 43 | Montreal Pierre Elliot Trudeau Int'l A | 45.47 | 73.75 | 36 | 3,215,694 | 1991–2000 |
| 44 | Montreal Saint Hubert A | 45.52 | 73.43 | 27.4 | 3,215,694 | 1991–2000 |
| 45 | Quebec City A | 46.8 | 71.38 | 74.4 | 635,184 | 1996–2005 |
| 46 | Fredericton A | 45.87 | 66.51 | 20.7 | 54,068 | 1991–2000 |
| 47 | Sept-Iles A | 50.22 | 66.27 | 54.4 | 23,636 | 1991–2000 |
| 48 | Yarmouth A | 43.83 | 66.09 | 43 | 7561 | 1991–2000 |
| 49 | Saint John A | 45.32 | 65.83 | 108.8 | 90,762 | 1991–2000 |
| 50 | Bathurst A | 47.63 | 65.65 | 58.8 | 16,427 | 1994–2003 |
| 51 | Miramichi A | 47.01 | 65.47 | 32.9 | 17,537 | 1992–2001 |
| 52 | Moncton A | 46.1 | 64.68 | 70.7 | 90,359 | 1991–2000 |
| 53 | Gaspe A | 48.78 | 64.48 | 34.1 | 3277 | 1994–2003 |
| 54 | Churchill Falls A | 53.55 | 64.1 | 439.5 | 705 | 1981–1990 |
| 55 | Summerside A | 46.44 | 63.83 | 19.5 | 14,654 | 2000–2009 |

**Table 1.** *Cont.*

| Station # | Station Name | Latitude (N) | Longitude (W) | Elev. (m) | Population | Data Range |
|---|---|---|---|---|---|---|
| 56 | Halifax Stanfield A | 44.83 | 63.5 | 145.5 | 276,221 | 1991–2000 |
| 57 | Shearwater A | 44.63 | 63.5 | 44 | 276,221 | 1991–2000 |
| 58 | Charlottetown A | 46.28 | 63.13 | 48.8 | 38,114 | 1991–2000 |
| 59 | Iles-de-la-Madeleine A | 47.42 | 61.78 | 10.7 | 12,010 | 1991–2000 |
| 60 | Sydney A | 46.17 | 60.05 | 61.9 | 33,913 | 1991–2000 |
| 61 | Sable Island A | 43.93 | 60.01 | 5 | 5 | 1991–2000 |
| 62 | Gander A | 48.95 | 54.58 | 151.2 | 9391 | 1991–2000 |
| 63 | Argentia A | 47.3 | 54 | 15.5 | 3496 | 1976–1985 |
| 64 | St John's A | 47.62 | 52.74 | 140.5 | 122,709 | 1991–2000 |

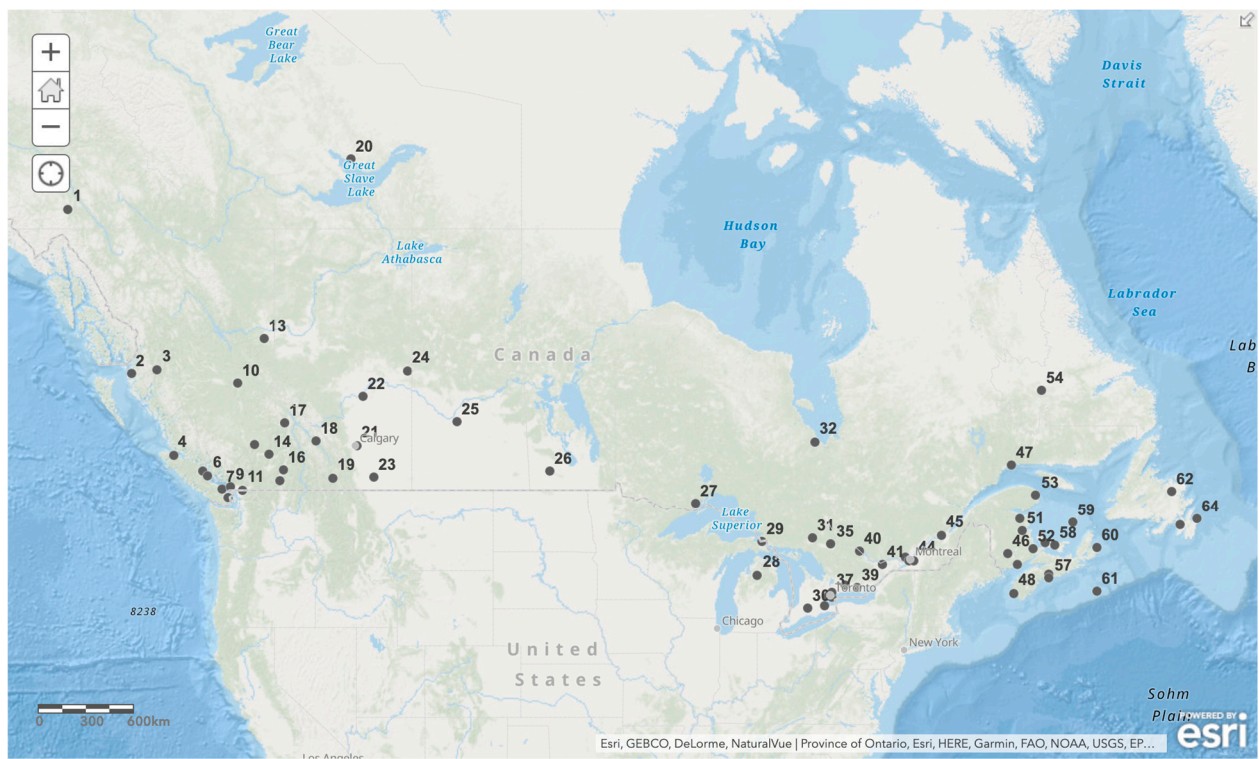

**Figure 1.** Map of all climate stations locations numbered as shown in Table 1.

Three existing tests were performed to determine the nature of the local airport environments. These were a test for peri-urban [9] using an R$\Delta$T$_{min}$ threshold of 1.05, rural/urban [12,13] using a $\Delta$DTD threshold of 0.2, and marine [14,15] using a threshold of ADTD$_{Tmin}$ of 2.35. A new test for a mountain environment was developed in this paper. We present the formalism for each of the metrics below. The DTD formalism was introduced by [11]:

$$DTD = \sum_{i=1}^{N} \frac{|T_i - T_{i-1}|}{N-1} \qquad (1)$$

where i is a counter over the period of interest (typically a month) with N − 1 pairs of values.

Ref. [12] developed this further by introducing $\Delta$DTD, the difference of DTD for the maximum temperature and minimum temperature of the day, and applied this effectively to urban and rural environments. They found urban environments had considerably higher values compared to nearby rural locations.

$$\Delta DTD = DTD_{Tmax} - DTD_{Tmin} \qquad (2)$$

Ref. [9] examined the day-to-day warm and cold transitions:

$$\Delta T+ = \frac{\sum_{i=1}^{n-1}(T_{i+1} - T_i)}{N+} \text{ if } (T_{i+1} - T_i) > 0; \; 0 \text{ if } (T_{i+1} - T_i) < 0 \tag{3}$$

$$\Delta T- = \frac{\sum_{i=1}^{n-1}(T_{i+1} - T_i)}{N-} \text{ if } (T_{i+1} - T_i) < 0; \; 0 \text{ if } (T_{i+1} - T_i) > 0 \tag{4}$$

$$R\Delta T = \frac{\Delta T+}{\Delta T-} \tag{5}$$

where $N+$ is the number of warm transitions and $N-$ is the number of cold transitions.

Ref. [9] found that $R\Delta T$ for minimum temperature of the day displayed distinct behaviour for peri-urban environments ($R\Delta T_{min} > 1.05$). Finally, refs. [14,15] examined DTD variables for marine locations and found that $DTD_{Tmin}$ was a clear indicator of such. One study [15] finetuned this by removing the winter months (December, January, and February) for Canadian locations ($ADTD_{Tmin}$).

Based on [9], it was hypothesized that most airport climate station data would be peri-urban in nature. We note that 52 of the 64 airport stations have been added to this study beyond those used in [9], all of which were deemed peri-urban. In [15], coastal environments dominated the thermal response and this interfacing would be explored in this paper. In addition, this study uncovered an additional thermally distinct regime, mountain environments, which tends to erase the peri-urban signal. To test if there was a distinct airport climate, we explored the relationship between urban size and the peri-urban metric. If a local airport climate was not generated, then lower population centres should have a weaker peri-urban signal.

## 3. Results

The airport locations are listed in Table 2 and ordered by $R\Delta T_{min}$, a metric used in [9] to identify peri-urban landscapes. Of the 64 airport locations, 55 (86%) had an $R\Delta T_{min}$ value of 1.05 or greater, the peri-urban threshold used to identify peri-urban landscapes in [9]. Most airports are situated at the interface between urban and rural landscapes and thus, a peri-urban identification was not surprising.

Nine locations did not meet this threshold. All of these stations are located in British Columbia, a western province characterized by both marine and mountain climates. In [9], marine (or coastal) climates were examined, and a metric was developed to detect marine environments, a winter-adjusted $DTD_{Tmin}$. A threshold that lies between 2.25 and 2.38 effectively differentiated between marine (less than threshold) and non-marine environments (greater than threshold). If we apply this criterion ($ADTD_{Tmin} < 2.35$) to the airport data, twenty were marine as listed in Table 3 with $ADTD_{Tmin}$, $DTD_{Tmin}$, and $R\Delta T_{min}$, and of these twenty, three fell below the peri-urban threshold. These were Abbotsford A, Nanaimo A, and Campbell River A. All three are from British Columbia. Abbotsford A is located in the Fraser River Valley. Nanaimo and Campbell River are coastal communities of Vancouver Island. For these three, the marine climate dominates the local conditions which effectively erases the peri-urban climate that typically forms for airport locations. These account for three of the nine locations in Table 2 that fell below the peri-urban threshold. What of the other six?

**Table 2.** List of airport climate stations. The stations are ordered from largest to smallest according to $R\Delta T_{min}$, a metric used to identify peri-urban landscapes in [9].

| Airport | $R\Delta T_{min}$ | Airport | $R\Delta T_{min}$ |
|---|---|---|---|
| Gaspe A | 1.34 | Calgary A | 1.14 |
| Churchill Falls A | 1.32 | St John's A | 1.14 |
| Moosonee A | 1.32 | Ottawa A | 1.13 |
| Moncton A | 1.30 | Cold Lake A | 1.12 |

**Table 2.** *Cont.*

| Airport | RΔT$_{min}$ | Airport | RΔT$_{min}$ |
|---|---|---|---|
| Saint John A | 1.29 | North Bay A | 1.12 |
| Fredericton A | 1.28 | Prince Rupert A | 1.12 |
| Shearwater A | 1.27 | Sable Island A | 1.12 |
| Toronto Buttonville A | 1.26 | Saskatoon A | 1.12 |
| Halifax Stanfield A | 1.26 | Argentia A | 1.11 |
| London A | 1.26 | Bathurst A | 1.11 |
| Trenton A | 1.25 | Edmonton Int'l A | 1.11 |
| Yarmouth A | 1.25 | Terrace A | 1.11 |
| Montreal Mirabel Int'l A | 1.24 | Whitehorse A | 1.11 |
| Petawawa A | 1.24 | Sudbury A | 1.11 |
| Montreal Saint Hubert A | 1.24 | Iles-de-la-Madeleine A | 1.10 |
| Thunder Bay A | 1.24 | Toronto Island A | 1.09 |
| Toronto Pearson Int'l A | 1.24 | Victoria A | 1.08 |
| Peterborough A | 1.23 | Yellowknife A | 1.08 |
| Gander A | 1.20 | Prince George A | 1.08 |
| Port Hardy A | 1.20 | Comox A | 1.07 |
| Quebec City A | 1.20 | Clinton A | 1.07 |
| Sault Ste Marie A | 1.20 | Vancouver Int'l A | 1.05 |
| Sept-Iles A | 1.20 | Dawson Creek A | 1.05 |
| Charlottetown A | 1.19 | Abbotsford A | 1.04 |
| Lethbridge A | 1.19 | Penticton A | 1.03 |
| Sydney A | 1.19 | Kamloops A | 1.02 |
| Wawa A | 1.21 | Nanaimo A | 1.02 |
| Summerside A | 1.18 | Cranbrook A | 1.00 |
| Hamilton A | 1.17 | Campbell River A | 0.99 |
| Montreal Pierre Elliot Trudeau Int'l A | 1.17 | Blue River A | 0.97 |
| Miramichi A | 1.16 | Golden A | 0.97 |
| Brandon A | 1.14 | Kelowna A | 0.95 |

**Table 3.** List of airport stations that met the marine threshold from [15] Gough. The stations are ordered by RΔT$_{min}$, the peri-urban threshold used in Table 2. Included in the table are elevation, ADTD$_{Tmin}$, and DTD$_{Tmin}$.

| Marine Stations | Elevation (m) | ADTD$_{Tmin}$ | DTD$_{Tmin}$ | RΔT$_{min}$ |
|---|---|---|---|---|
| Shearwater A | 44.0 | 1.95 | 2.3 | 1.27 |
| Halifax Stanfield A | 145.5 | 2.13 | 2.49 | 1.26 |
| Yarmouth A | 43.0 | 2.06 | 2.44 | 1.25 |
| Port Hardy A | 21.6 | 1.90 | 1.92 | 1.20 |
| Gander A | 151.2 | 2.15 | 2.42 | 1.20 |
| Sydney A | 61.9 | 2.35 | 2.67 | 1.19 |
| Summerside A | 19.5 | 2.32 | 2.63 | 1.18 |
| St John's A | 140.5 | 2.12 | 2.38 | 1.14 |
| Sable Island A | 5.0 | 1.69 | 1.97 | 1.12 |
| Prince Rupert A | 35.4 | 2.09 | 2.17 | 1.12 |
| Argentia A | 15.5 | 1.57 | 1.89 | 1.11 |
| Terrace A | 217.3 | 1.62 | 1.63 | 1.11 |
| Iles-de-la-Madeleine A | 10.7 | 1.88 | 2.16 | 1.10 |
| Toronto Island A | 76.5 | 1.94 | 2.28 | 1.09 |
| Victoria A | 19.5 | 1.80 | 1.84 | 1.08 |
| Comox A | 25.6 | 1.84 | 1.89 | 1.07 |
| Vancouver Int'l A | 4.3 | 1.71 | 1.7 | 1.05 |
| Abbotsford A | 59.1 | 1.94 | 1.96 | 1.04 |
| Nanaimo A | 28.0 | 2.23 | 2.24 | 1.02 |
| Campbell River A | 108.8 | 2.31 | 2.31 | 0.99 |

These six are located in the mountainous interior of British Columbia (Table 4). To date, within the DTD temperature framework, a threshold for mountain climates has not been developed. Thus, this provided an opportunity to do so. In examining the existing metrics, all six in Table 4 were identified as rural using $\Delta$DTD but the magnitudes of $DTD_{Tmin}$ and $DTD_{Tmax}$ were less than is typical for a rural site. In Table 3, these remaining stations are listed with $\Delta$DTD and $DTD_{Tmin}$, noting that all these stations have a $R\Delta T_{min}$ less than 1.05. All stations have $\Delta$DTD below 0.2, thus achieving a rural designation [9,12]. All stations have a $DTD_{Tmin}$ below 3.15 but none of these stations meet the marine requirement; thus, these mountain stations occupy the range of $DTD_{Tmin}$ that exceeds the tightly constrained marine environments [15] but is more constrained than rural environments [9,12,13].

**Table 4.** Mountain airport locations.

| Mountain Stations | Elevation (m) | $DTD_{Tmin}$ | $\Delta$DTD | $R\Delta T_{min}$ |
|---|---|---|---|---|
| Penticton A | 344.4 | 2.75 | −0.51 | 1.03 |
| Kamloops A | 345.3 | 2.51 | 0.16 | 1.02 |
| Cranbrook A | 940.0 | 2.71 | −0.01 | 1.00 |
| Blue River A | 690.4 | 3.13 | −0.22 | 0.97 |
| Golden A | 784.1 | 2.65 | −0.21 | 0.97 |
| Kelowna A | 429.5 | 2.67 | −0.21 | 0.95 |

Finally, we addressed the research question: do airports, in and of themselves, generate an "airport" climate? To explore this, we conducted a correlation analysis (linear regression) of the population with the $R\Delta T_{min}$ metric. This is displayed in Figure 2. The logarithm of population was used as has been done in other studies [17]. The airport on Sable Island with a population of five was not included in this analysis as it is a national park reserve and has restricted access.

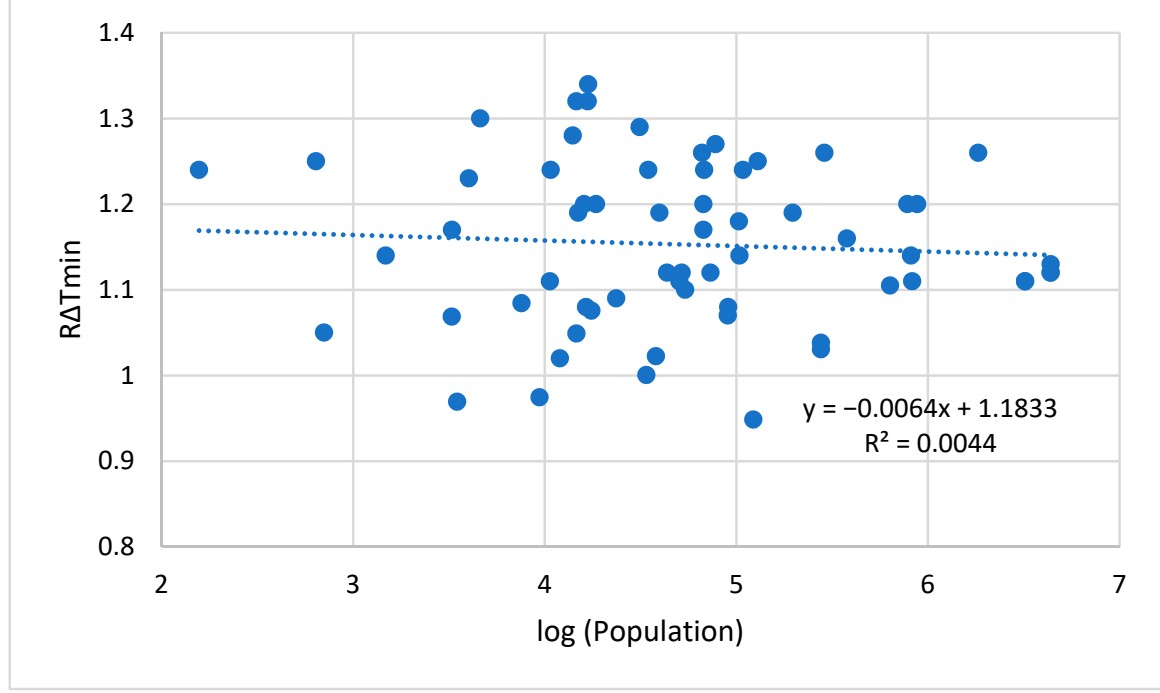

**Figure 2.** Plot of log(population) versus $R\Delta T_{min}$ for sixty-three airport stations. Sable Island with virtually no population was not included.

The correlation (r ≈ 0.066) was extremely low, indicating that there was not a tendency for lower population stations to have a weaker peri-urban signal and vice versa. This strongly suggests that the structure of the airport generates a localized peri-urban

environment. If we ranked the data by $R\Delta T_{min}$ values and compared the top ten with the bottom ten population (LogP) values, the two groups were not significantly different using the Student's *t*-test. This was especially true if the locations below the 1.05 threshold were not used. Thus, extremely large values of $R\Delta T_{min}$ were not associated with high population centres and vice-versa.

## 4. Discussion

As anticipated, most (86%) of the airport climate data stations examined reported a peri-urban climate using the $R\Delta T_{min}$ threshold of 1.05 developed by [9]. Of the remaining 14%, three stations were identified as marine using criteria for $ADTD_{Tmin}$ below 2.35 developed by [15]. The final six stations were all from mountainous regions of British Columbia and a new category of "mountain" was proposed using a two-step identification of $\Delta DTD$ below 0.2 [9,12] and $DTD_{Tmin}$ below 3.20.

Ref. [9] provided a physical mechanism for the behaviour of the $R\Delta T_{min}$ metric for peri-urban environments. Temperature transitions, more generally, arise from both energy balance (radiatively driven) and dynamic changes (advection) in the atmosphere. The diurnal cycle of temperature largely arises from the daily energy balance cycle with solar energy as the dominant input of energy to a location that is modified by surface conditions (albedo, heat storage) and atmosphere conditions (radiatively active gases, cloud and fog coverage). The atmosphere flows and thermal changes occur at a range of spatial scales for Canadian latitudes due to shifting air masses, the episodic presence of midlatitude cyclones, convective storms, and sea and lake breezes that characterize many areas of southern and northern Canada [18,19]. Given the distinctive characteristics of peri-urban environments, localized radiative and dynamic features that occur at the boundary of the urban–rural interface explains the thermal transition behaviour. As the $R\Delta T_{min}$ metric indicated, the warm transitions were larger in magnitude than cold transitions for the peri-urban environments. Cloud and fog cover can explain some of the radiative changes experienced in peri-urban landscapes. Cloud and fog cover and the increased absolute humidity associated with them serve to trap outgoing radiation, thus enhancing warm transitions and mitigating cold transitions. The country breeze arises from the horizontal temperature gradient that forms between rural and urbanized landscapes. The less-dense air over the urbanized landscape rises and thus draws in air from rural areas that tend to be moister. This type of circulation is conducive for the formation of low-lying clouds. However, if this breeze and cloud and fog formation were to occur every night, the impact on DTD temperature variation would be negligible and the $T_{min}$ transition from day-to-day would be minimized, not enhanced. Not every night, though, does a country breeze develop, especially when regional or larger-scale winds dominate the local weather conditions. Additionally, not every country breeze induces the instability necessary for the production of condensation. It is the accumulated variability of these mechanisms that enables the $T_{min}$ transitions to be larger than in rural or urban settings [9]. This is consistent with the general urban heat island effect that is largely driven by temperature inversion on a mesoscale at higher altitudes in the atmosphere during the daytime and on a microscale at lower altitude during nighttime [20]. The airport was storing the heat from solar radiation during the day and releasing them at night at the surface level of the atmosphere, which led to a reduction of $T_{min}$ transition at the airport compared to a rural site.

Airports tend to be located at the fringe of urban communities, in peri-urban regions. Some airports are located in rural areas but still display peri-urban characteristics. Thus, it appears that airports are peri-urban environments in and of themselves, thus generating a microclimate or "airport" climate [21]. Airports are largely stripped of ground vegetation and most often have paved runways that mimic urban ground cover. Given the domination of airports in the identification of peri-urban environments in [9], is the "peri-urban" climate better characterized as an "airport" climate? In [9], as noted above, the impact of the development of the urban area of Toronto was examined in a longitudinal study starting in the 1840s when the University of Toronto climate station was located on the

University lands north of the burgeoning urban center. The climate data indicated a rural signature at that time. While the location of the University did not change, the city grew around it. From the 1870s to the 1890s, a distinct peri-urban signal was detected without the presence of an airport, which transitioned to urban after this time as the city grew even larger. Since meteorological data are key for the operation of airports, it is not surprising that most of the station data currently showing peri-urban characteristics are at airports.

The peri-urban climate with the exception of a few marine and mountain locations was found at all other airports (Table 2), and these airports were located at a wide range of population centres varying from 5 to over 2 million (Table 1). A statistically significant relationship between the extent of urbanization, using population as the proxy variable, and the strength of the peri-urban metric was not found (Figure 2). This indicated the air-port itself generates the peri-urban climate even when the surrounding environment can be classified as rural. Given that rural airports are stripped of vegetation with extensive tarmac for runways and typically surrounded by rural settings, the generation of a peri-urban climate is a reasonable explanation. This notion of an airport climate should be further explored and quantified. Since airport climate records are often used as a representation of an urban environment, such use may need to be nuanced [19].

Nine airports did not have a $R\Delta T_{min}$ value of greater than 1.05. Three of them were characterized as marine (Abbotsford A, Nanaimo A, and Campbell River A), part of a larger group of 20 of the 64 stations that were marine in nature. The other seventeen had $R\Delta T_{min}$ that exceeded 1.05. However, this group as a cohort had lower $R\Delta T_{min}$ values than the remainder of those above 1.05 (mean = 1.09 for marine, 1.19 for other peri-urban) in a statistically significant fashion using the Student's *t*-test ($p < 0.001$), indicating distinct, although overlapping, populations. The marine climate tended to mitigate extremes as seen in the DTD metrics with the lowest values of these of all climate types (urban, rural, marine, and peri-urban). Thus, the peri-urban warm extremes exemplified in the $R\Delta T_{min}$ metric were dampened, likely due to the more pervasive presence of fog and low-lying cloud cover which is more episodic in peri-urban climates further inland [14,15]. Thus, the marine climate acted to mitigate and in some cases erase the peri-urban signal as seen in the three locations that dropped below 1.05 for $R\Delta T_{min}$.

The six remaining stations that did not meet the $R\Delta T_{min}$ 1.05 threshold were all located in mountainous regions of British Columbia. Mountainous regions in different parts of the world develop distinctive diurnal circulations [22–24]. Typically, during the day, upslope winds develop with descending air interior to the valley walls and vice versa at night, i.e., anabatic and katabatic winds, respectively. Furthermore, when prevailing air flow is parallel to the mountains, it can funnel winds into a certain direction due to the orographic effects of the mountains [2]. This strong diurnality, not unlike the marine environments discussed above, tends to dominate the air temperature distribution, effectively minimizing other effects such as urbanization of the land surface. The six stations all were below the peri-urban threshold using $R\Delta T_{min}$ and were also deemed "rural" using the $\Delta$DTD metric. However, unlike non-mountainous rural locations, the values of $DTD_{Tmin}$ and $DTD_{Tmax}$ were muted, although not as muted as they were for marine environments. Thus, it appears that the presence of the well-established diurnal circulation acted to erase the urban signature for airports in mountain areas and dampen the rural amplitude. Therefore, additional care should be taken when analyzing the urban heat island effects in mountainous areas, particularly if one or more airport station were used to represent urban or rural setting. The urban heat island signature may be dampened by the airport environment in mountainous regions and can lead to underestimation in urban heat island studies in cities such as Ulaanbaatar, Mongolia [25] and Cluj-Napoca, Romania [24].

From the set of 64 airports examined in this study, a number of thresholds for the various thermal metrics could be used to sort through the stations. The metrics included $R\Delta T_{min}$ (peri-urban), $\Delta$DTD (urban and rural), $ADTD_{Tmin}$ (marine), and $DTD_{Tmin}$ (mountain). The following decision flowchart could be followed (Figure 3) and the results as applied to the sixty-four climate stations depicted in Figure 4. This figure illustrates the

distinctive nature of the mountainous climates, the new category developed in this study. The above examples indicate that the airport itself generates a peri-urban climate even when the surrounding environment is classified as rural, marine, or mountain.

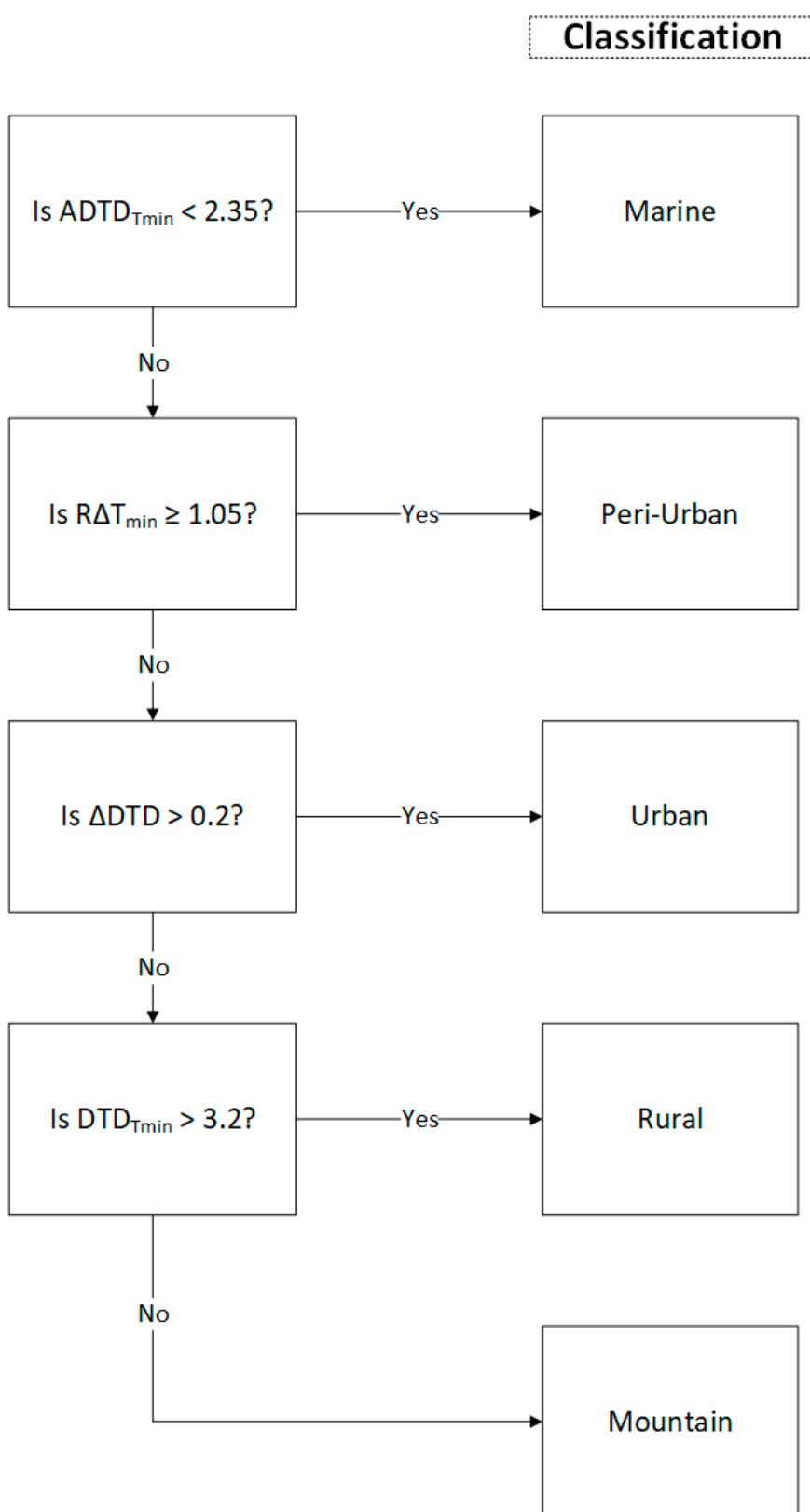

**Figure 3.** Decision flowchart to determine the nature of climate station environment.

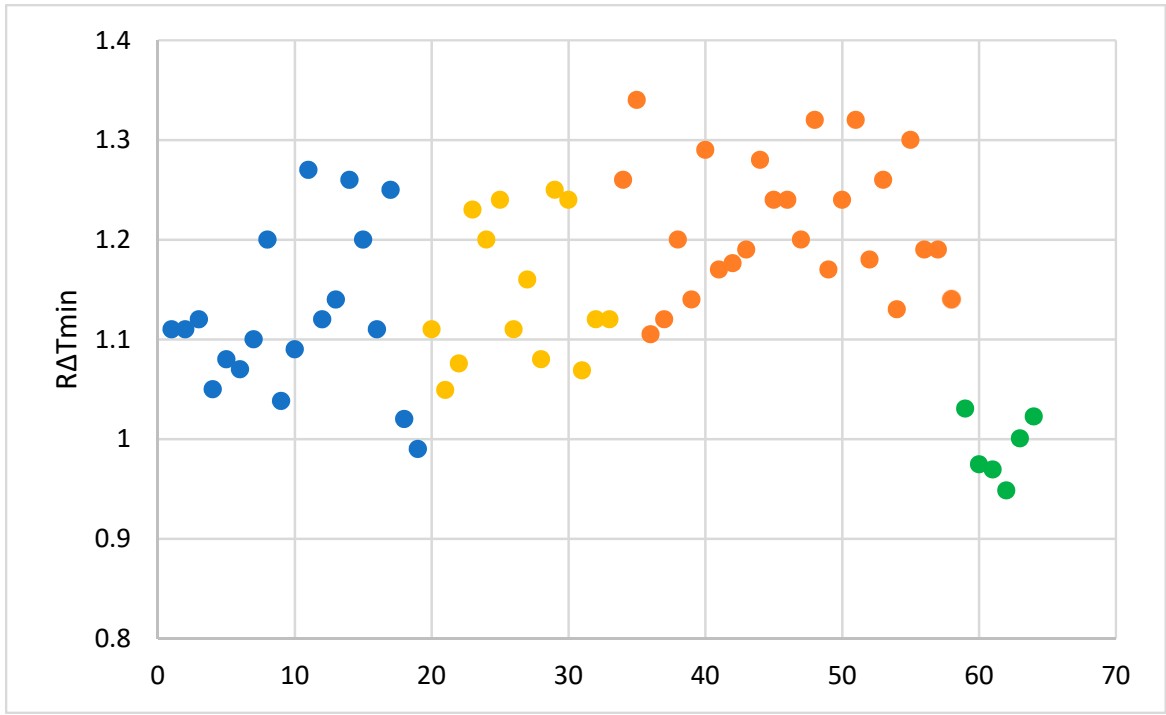

**Figure 4.** Stations binned by marine (blue), peri-urban with a secondary identification as rural (light orange), peri-urban with a secondary identification as urban (dark orange), and mountain (green), plotted as a function of $R\Delta T_{min}$ following the flow chart in Figure 3.

## 5. Conclusions

In this study, we explored the nature of climate records at sixty-four Canadian airports. Most (86%) of these records were identified as peri-urban consistent with [9]. Of the remainder, three were unambiguously identified as marine using criteria developed in [15]. The final six climate records were identified using newly developed criteria as "mountain", reflective of their location in the mountainous region of British Columbia and reflective of a well-defined diurnal circulation. As a result of this, a flow chart was developed using four DTD temperature variability metrics to correctly identify the local environment of the climate stations: marine, peri-urban, urban, rural, and mountain. This paper provided evidence of a specialized "airport climate", a well-defined microclimate that is generated by the nature of an airport and its modification of the local environment as originally suggested in [21]. It also revealed the dampening effect on the urban heat island signature in mountainous regions. The airport climate needs further exploration in other parts of the world as well as for other thermal metrics.

**Author Contributions:** Conceptualization, W.A.G. and A.C.W.L.; methodology, W.A.G.; formal analysis, W.A.G.; data curation, W.A.G.; writing—original draft preparation, W.A.G. and A.C.W.L.; writing—review and editing, A.C.W.L. and W.A.G.; visualization, W.A.G. and A.C.W.L.; funding acquisition, W.A.G. All authors have read and agreed to the published version of the manuscript.

**Funding:** This research was funded by Natural Sciences and Engineering Research Council of Canada (NSERC), grant number RGPIN-2018-06801.

**Institutional Review Board Statement:** Not applicable.

**Informed Consent Statement:** Not applicable.

**Data Availability Statement:** The data presented in this study are available from https://climate.weather.gc.ca/ (accessed on 8 February 2022).

**Conflicts of Interest:** The authors declare no conflict of interest.

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
