# Peer review of "Do Airports Have Their Own Climate?"

_2674-0494, doi:10.3390/meteorology1020012_

Round 1
Reviewer 1 Report
The study expands on previous similar studies and examines the thermal climate of 64 airports across Canada. The study supports previous work in that airports are typically classed as peri-urban, which combined with the general location of most airports outside of population centres, may suggest airports themselves are responsible for this climate. The study is interesting, but could be drastically improved with some additional clarity around testing procedures and thresholds used to define climates. Specific comments below.
Line 35 – “environment” should be “environments”
Line 36 – “compared” should be “comparing”
Line 76 - “This” should read “The current work” or similar
Line 85 – I find the discussion on the three tests difficult to follow. You need to clearly lay out what tests you are going to perform, and what criteria will then be used to define climates. It should be clearly stated at the outset, for example, you are looking for a 1.05 threshold for a peri-urban environment. This is listed on line 105, but in brackets, and it wasn’t until the results section this became clear to me this is the metric you are using. You very neatly lay out a decision flow chart in figure 3, something similar in this materials and methods discussion to allow the reader to follow your study would be very useful.
Line 89 – I assume DTD is “day to day”, but this should be defined the first time it is used
Line 104 to 107 – referencing needs to be consistent throughout, in these two sentences you have used number only, number with name, and name only.
Line 110 = “airports” should be “airport”
Line 111 – you state 12 airports are analysed in Gough 2020, while on line 76 you state 13?
Line 134 – “climates” shouldn’t be in brackets, it should read “marine (or coastal) climates
Line 133 to 136 – this sentence is very long and hard to understand
Line 149 – I think you mean table 4?
Line 220/221 - “climates” should be “climate”
Line 239/240 – sentence does not make sense
Line 293 – your concluding sentence needs to be expanded, first by defining “this” as an “airport climate”, and secondly by suggesting ways in which this should be further explored.
Author Response
Comments and Suggestions for Authors
The study expands on previous similar studies and examines the thermal climate of 64 airports across Canada. The study supports previous work in that airports are typically classed as peri-urban, which combined with the general location of most airports outside of population centres, may suggest airports themselves are responsible for this climate. The study is interesting, but could be drastically improved with some additional clarity around testing procedures and thresholds used to define climates. Specific comments below.
Line 35 – “environment” should be “environments”
Done
Line 36 – “compared” should be “comparing”
Done
Line 76 - “This” should read “The current work” or similar
Revised as suggested
Line 85 – I find the discussion on the three tests difficult to follow. You need to clearly lay out what tests you are going to perform, and what criteria will then be used to define climates. It should be clearly stated at the outset, for example, you are looking for a 1.05 threshold for a peri-urban environment. This is listed on line 105, but in brackets, and it wasn’t until the results section this became clear to me this is the metric you are using. You very neatly lay out a decision flow chart in figure 3, something similar in this materials and methods discussion to allow the reader to follow your study would be very useful.
This is a good point. For clarity, the thresholds for the peri-urban, urban/rural and marine have been added into the Materials and Method section.
Line 89 – I assume DTD is “day to day”, but this should be defined the first time it is used
Thank you. We have defined its first usage on line 53 and in the abstract.
Line 104 to 107 – referencing needs to be consistent throughout, in these two sentences you have used number only, number with name, and name only.
We have addressed the referencing throughout the document.
Line 110 = “airports” should be “airport”
Corrected
Line 111 – you state 12 airports are analysed in Gough 2020, while on line 76 you state 13?
Apologies for the confusion. 12 of the 64 were used in Gough (2020). This has been corrected.
Line 134 – “climates” shouldn’t be in brackets, it should read “marine (or coastal) climates
Corrected
Line 133 to 136 – this sentence is very long and hard to understand
For clarity, this sentence has been broken into two sentences.
Line 149 – I think you mean table 4?
Correct. We have added that to clarify
Line 220/221 - “climates” should be “climate”
Corrected. Thank you
Line 239/240 – sentence does not make sense
Good catch. There was a missing word, “environment” has been added after “urban”. We have also included a reference to Zaknic-Catovic and Gough (2021) [citation 19]
Line 293 – your concluding sentence needs to be expanded, first by defining “this” as an “airport climate”, and secondly by suggesting ways in which this should be further explored.
We have indicated these results should be explored for other geographic locations. This work narrowly focuses on day-to-day temperature variability and thus a more thorough exploration of the airport climate, now that such has been identified, should be done. These two points have been added.
Reviewer 2 Report
Suggestions:
- add more relevant references, especially to the discussion chapter
- please state the limitations of the study (e.g. period used 1991-2000, why not later years?)
- what about the transferability of the method to other regions of the world?
Author Response
add more relevant references, especially to the discussion chapter
Zaknic-Catovic and Gough (2021), Ningram (2018), Herbel et al. (2016), Ganbat et al. (2013) have been added to the Discussion.
please state the limitations of the study (e.g. period used 1991-2000, why not later years?)
Starting from 2011, the ownership, operation and maintenance of staffed airport weather stations in Canada were transferred to a different organization (NavCan). Due to these changes, we are uncertain if this led to non-climatic shifts in the data. Furthermore, the amount of work done on quality control on airport weather data declined over time. In addition, choosing the 1991-2000 period would remove some of the influence from climate change, thus reducing the impact from this factor while focusing more specifically on comparing urban, peri-urban, coastal, mountain and rural environments.
The use of 1991-2000 is consistent with Gough (2020) and Gough (2022) as now noted.
what about the transferability of the method to other regions of the world?
Thank you for your suggestion. We have added examples from mountainous regions where urban heat island signatures from Mongolia (Ganbat et al., 2013) and Romania (Herbel et al., 2016) would become underestimated due to the usage of airport weather stations in their analysis.
Reviewer 3 Report
Review of the manuscript "Do airports have their own climate?", by William A. Gough and Andrew C.W. Leung
As an important part of air navigation services at airports meteorological service relies on high grade instruments providing reliable and consistent information about the weather that allows studying local climate. Usually, airports are located at urban fringe so it interesting to evaluate their “thermal signature” and understand if it results of airports proximity to cities or large impermeable surfaces with high thermal inertia intrinsic to airports make airport climate to resemble peri-urban climate. The presented work demonstrates that there is no statistically significant relationship between the used peri-urban metric and the population of the nearby town. The latter is interpreted by the authors as evidence that airports generate their own local climate. The authors also prove that the peri-urban signal can be significantly suppressed in coastal or mountainous environment.
Given the above and in the context of climate variability I believe the presented results are of interest to the scientific community and deserve to be published. The manuscript looks fine, and I have no major comments. As a minor comment the manuscript only needs to be consistent with the journal citation convention.
Author Response
Thank you for your review. We have adjusted the citation format to be compliant with journal’s preferred style.